# Analyzing the Concept of Corporate Sustainability in the Context of Sustainable Business Development in the Mining Sector with Elements of Circular Economy

**Ekaterina Blinova [1],\*, Tatyana Ponomarenko [1] and Valentin Knysh [2]**

1    Faculty of Economics, Saint Petersburg Mining University, 199106 Saint Petersburg, Russia; ponomarenko_tv@pers.spmi.ru
2    Luzin Institute for Economic Studies—Subdivision of the Federal Research Centre, Kola Science Centre of the Russian Academy of Sciences, 184209 Apatity, Russia; knyshva@mail.ru
\*    Correspondence: s205085@stud.spmi.ru

**Abstract:** Promoting the concept and principles of sustainable development at the micro level requires that industrial companies understand and improve approaches to managing corporate sustainability. Currently, economics does not provide a universal definition of what corporate sustainability is. With regard to the mining sector, corporate sustainability issues reflecting the viability, value, and sustainable development potential of companies have not yet been studied extensively. The article discusses the conceptual foundations of corporate sustainability; the characteristics and a classification of approaches to defining corporate sustainability; and the relationship between corporate sustainability, sustainable development at the micro level, and circular economy. By analyzing the example of Russia, the influence of the mining industry on the environmental, economic, and social development of both a country with a resource-based economy and individual mining regions is shown from the viewpoint of sustainability. The distinguishing features of mining companies, which include natural capital and mineral assets, are studied in the context of promoting corporate sustainability. It is proven that the effective corporate management of ESG factors results in environmental and social influence that goes in line with sustainable development requirements and serves as a foundation for corporate sustainability. A refined definition of corporate sustainability has been formulated, the specific features of corporate sustainability management in mining companies have been determined, and the specific features of corporate social responsibility have been studied. The issue of integrating circular economy elements into the corporate sustainability concept is discussed, and it is claimed that the inclusion of circular business models in the corporate strategies of mining companies will contribute to their corporate sustainable development and boost their contribution to the achievement of sustainable development goals.

**Keywords:** corporate sustainability; sustainable development; corporate sustainability in the mining sector; corporate social responsibility; circular economy

## 1. Introduction

Over the past few years, sustainable development (SD) issues have become the subject of many discussions revolving around the problem of adjusting and achieving SD goals at the national, regional, sectoral, and corporate levels, including in the mining industry. The growth of interest in this issue is confirmed by a large number of academic publications [1–3], analytical reviews and studies [4–6], and international initiatives [7–10].

Research in this area is needed for many theoretical and practical reasons. The SD theory has not yet solved problems associated with SD evaluation at various levels or with the definition of SD. Moreover, the relationships between different evaluation indicators

have not been established, the choice of key SD drivers has not been substantiated, and management tools have not been sufficiently developed [11]. In practical terms, the requirements for sustainable reporting have not been harmonized, and the best practice of applying SD management tools, including resource and environmental policy, low-carbon economy, waste management, and best available technology (BAT) application, is yet to be developed [12]. At the same time, the last decade has seen several long-term environmental, economic, and social trends: the strengthening of the influence of civil initiatives, the growing urgency of environmental problems, the use of renewable energy sources, improvements in power generation efficiency, and the emergence of new global risks and challenges. All this necessitates a search for models of socioeconomic development at various levels with a focus on sustainable development [13–15].

Achieving the UN's Sustainable Development Goals (SDGs) related to resource efficiency, environmental impact, and human well-being depends on which consumption and production models will dominate the economic practices conducted by countries, industrial sectors, and companies, including those engaged in mining. The transition to sustainable consumption and production patterns is the focus of SDG 12, with the goal of achieving sustainable management and efficient use of natural resources by 2030 [16,17], separating economic growth from the consumption of primary resources and environmental degradation. Decoupling effects can be achieved by moving from the current linear economy model to a low-carbon circular economy [18,19].

Circularity is increasingly beginning to be seen as a way to achieve sustainable consumption and production in relation to other SDGs [18,20]. This approach makes the concept of circular economy (CE) relevant in the context of solving corporate sustainability (CS) issues and assessing the contribution of companies to the achievement of the SDGs that are related to the efficient use of limited natural resources, including mineral resources. Although the transition to a CE is generally consistent with the ideas of sustainability and the goals of SD with both SD ideas and sustainability goals [18,21], the interdependence between sustainability and CE is not always clear due to differences in how CE is conceptualized [22,23]. This makes it difficult to develop and standardize micro-level CE indicators and models for their assessment [23] and creates barriers to CE ideas in corporate governance. While mining companies have a great potential for circularity, an additional constraint is the lack of institutional incentives when the national or international regulator does not include the mining sector in the list of priority sectors for transitioning to CE and monitoring this process. Nonetheless, some studies show that mining companies can make great progress if they apply CE principles [24], while failure to implement CE practices puts mining companies at risk of falling behind in innovative business practices. For example, it has been found that in China, companies feel a lack of integration with sustainable development paradigms such as CE, but if large enterprises are able to implement effective and sustainable practices, then small and medium-sized enterprises do not seek to implement 3R strategies in waste management [25].

The generally accepted idea of sustainable development is the definition given by the Brundtland Commission [26]. The current state of the concept of SD is characterized by ongoing discussions about the principles and factors of SD. They include the question of whether the SD concept is applicable to individual regions, industries, or companies; arguments concerning the place of the SD concept among other concepts related to CSR and CS issues; and how SD should be understood at the level of mining companies [27,28].

Considerable attention has been paid in the last decade to various aspects of SD assessment at the macro and micro levels and to the requirements for SD and ESG reporting at the international level [29–33].

At the same time, the issues of SD regulation at the industrial level remain poorly managed even for industries with a strong social and environmental impact, including the mining industry. The most well-known industry-specific standards are the SASB standards developed in 2018. They cover 77 industries, including metallurgy, oil and gas production, mining, and others.

Mining is associated with aa strong negative impact on the natural environment, which can take the following forms: soil disturbance, waste rock disposal, water pollution and drainage, landscape disfigurement, harmful emissions, water pollution by industrial effluents, flora and fauna deterioration, and other negative effects on ecosystems [34].

The degree of impact on the surrounding ecosystems depends on the extraction method, the type of minerals being mined, the technologies used, and other production factors in the extraction, processing, and use of mineral resources [35]. For example, in the Kansk–Achinsk coal basin in Russia, significant amounts of brown coal for power stations are mined by open pit mining. The average dust load is 200 to 700 t/km², with the maximum reaching 2000 t/km² per year. The specific soil capacity is 2 to 7 hectares per one million tons of coal, and, as a rule, the fertile soil layer is disturbed. Excavations extend for 30 km. As a result of drainage operations, a large amount of groundwater is pumped out: the specific water discharge is 0.2 to 0.6 m³/t of coal in large open pits and 1.5 to 30 m³/t in small open pits. The total drainage water discharge from the open pits grew by a factor of 1.5 from 1990 to 2010 [36].

Pollution is growing faster, outpacing the production growth rate for the most important types of minerals. The problem of the rapid growth of the negative impacts that mining companies have on the environment is illustrated by the following case in the coal industry. According to a report by Rosprirodnadzor on the Russian coal industry, all environmental indicators of the industry deteriorated from 2012 to 2018 (Figure 1), outpacing the 30% increase in coal production.

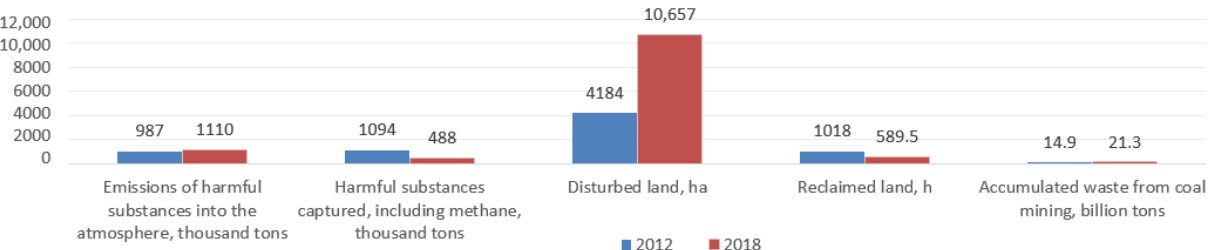

**Figure 1.** Environmental indicators in the Russian coal industry for the years 2012 to 2018 (developed by the authors based on [37]).

The data in Figure 1 show that emissions of harmful substances increased by 12%, the total area of disturbed land grew by 154%, and the volume of accumulated waste increased by 43%, while the amount of harmful substances captured fell by 55%, and the total area of reclaimed land decreased by 42%.

From 2009 to 2016, there was an increase in greenhouse gas emissions produced by Russia's mining and metallurgy sectors [38]: $CO_2$ grew by 13.7%, $CH_4$ grew by 15.8%, PFC grew by 8.6%, and $CF_4$ grew by 11.1%. $C_2F_6$ emissions from aluminum production decreased by 11.3%. During the period under consideration [39], iron ore production grew by 3.8%, chromium ore production grew by 32.5%, and iron ore raw material production increased by 23.8% [40]. Deloitte forecasts that from 2021 to 2026, greenhouse gas emissions produced by Russian steel companies will decrease [41]. In the period from 2016 to 2019 [42], waste from the extraction of ores used in metallurgy increased by 70%, and waste from metal production grew by 18.5%.

There are three key areas affected by the mining industry: employment, the environment, and social conditions [43]. The impact of the mining industry on employment is difficult to assess since the growth in automation slows down new employment. Local communities are often very concerned about environmental pollution and degradation and the industry's negative influence on the cultural environment [36]. There are also conflicts regarding the preservation of the local landscape, the traditional way of life, and the resettlement of the population for the development of natural resources. Currently, when a new deposit begins development, the company's employees work on a fly-in-fly-out

basis, its social infrastructure is created only for the staff, and the local population is not granted access to it. The profound influence of the mining industry on the socioeconomic development of the territories where it operates is an issue for the corporate sustainability of mining companies.

The goal of this study is to clarify the concept, place, and role of corporate sustainability in the context of sustainable development in the mining industry. The objectives of the study are: a comparative analysis of CS interpretations based on scientific literature published from 1998 to 2017 and a study of the essence of CS and its relationship with SD and management concepts; identifying the influence of the specific features of the industry and global SD requirements on corporate sustainability in the mining sector by analyzing the relationship between the concepts of sustainable development and the circular economy, including in relation to mining companies.

## 2. Materials and Methods

The methodology of this study includes case studies, systems-oriented analysis, decomposition, and comparative analysis. The study is based on an extensive list of sources. We conducted a literature review covering five interrelated aspects (boxes 1–5 in Figure 2) to study the topic under discussion in a holistic way and produce the necessary result (box 6).

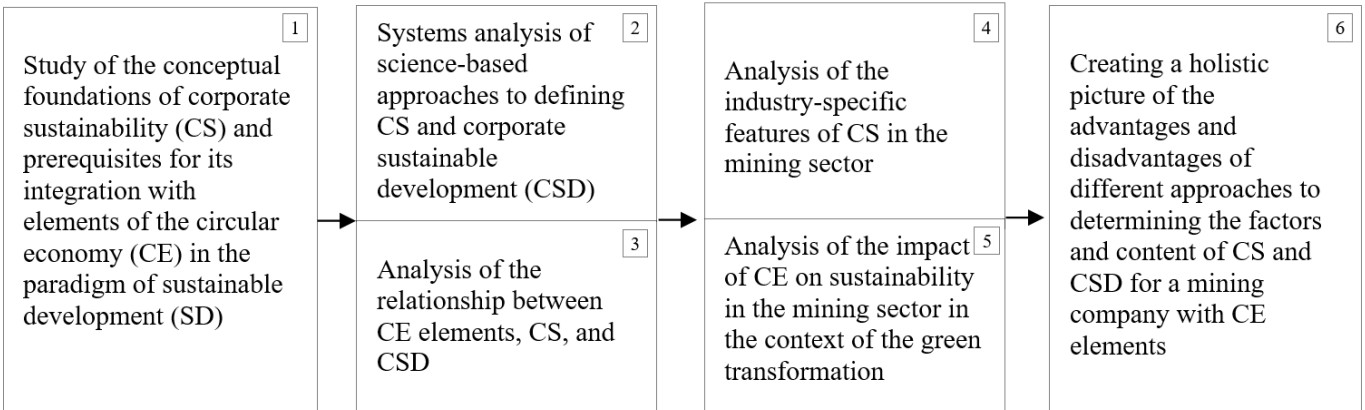

**Figure 2.** Theoretical foundations of the study.

We formulated the key research question: What are the current approaches to interpreting mining companies' corporate sustainability and sustainable development in the light of the transition to a circular economy?

We chose this methodology because of the need for a multidimensional consideration of the problem as it is related both to the sustainability of micro-level economic systems in the context of the SDGs and the sustainability of corporate structures such as mining companies that have industry-specific features and are difficult to adapt to CE models. The activity of mining companies is sensitive not only to market factors (price volatility, changes in demand) but also to noneconomic global challenges (climate change, regional conflicts). The movement towards CE models is problematic for the mining sector because it requires that companies switch to more rational and environmentally friendly mining practices while they have to deal with a decrease in the quality and availability of mineral resources and an increase in the volume of mine waste. In order to proactively respond to emerging threats and challenges and be ready for future changes in the external conditions, mining companies must follow new requirements and standards and update approaches to ensuring their sustainability. The research community can facilitate this process through studies that close the gaps between the already existing approaches to achieving CS and CSD and the current trends, such as moving towards CE practices, low-carbon development, and green transformation.

To broaden our understanding of CS, we use concepts connected with the idea of sustainability and discuss them in terms of their applicability to mining companies. We argue that studying management concepts (CSR, SD, ESG) makes it possible to identify the key CS and CSD elements for industrial companies, including mining companies. Additionally, studying the relationships between the concepts of CS and CE shows that they are not in conflict with each other, which allows us to claim that the integration between CS and GSD indicators and CE elements will be productive for mining companies.

The sources of information that we used are academic publications indexed in the Scopus and ISI Web of Science databases, UN reports, reports and analytical reviews produced by international (Deloitte, PWC) and Russian organizations in the field of sustainable development (the Analytical Center under the Government of the Russian Federation, the Federal Service for Supervision of Natural Resources of the Russian Federation), and the corporate reports of mining companies (SUEK, Metalloinvest).

## 3. Literature Review

For the first time, the concept of sustainable development applied at the company level appeared in the works by J. Elkington, M. Epstein and R. Steuer, which are now considered to be classics [44–47].

In the well-known 3P ("People, Planet, Profit") concept by J. Elkington, all elements are interconnected [44] and are a guideline for the company's operations, which projects the ideas of SD from the macro level to the micro level (the social, environmental, and economic development of the company).

"Sustainable business is the new managerial paradigm that Elkington presents for the next century. Business is sustainable when it lives up to the "triple bottom line" of economic prosperity, environmental quality and social justice. The three bottom lines are interrelated, interdependent, and partly in conflict" [48] (p. 229).

M. Epstein concluded that corporate sustainability is the result of activities in the field of sustainable development [45,46]. Companies should include sustainability steps in their business strategies that take into account the views of stakeholders and CSR.

R. Steuer defined corporate sustainability as a corporate strategy aimed at meeting the needs of the company and its stakeholders in order to preserve and develop the human and natural resources that it will need in the future [47]. Steuer's interpretation is based on the Brundtland definition applied at the company level. However, there is a distinction between SD and CS. SD is usually perceived as a social governance model that addresses a wide range of quality-of-life issues in the long term, while CS is a corporate governance model that takes into account the short- and long-term economic, social, and environmental performance of corporations. In our opinion, it is a quite logical theory, but it raises the problem of assessing the relationships between the needs of the company at the present time, the availability of resources for its operations, and the necessary resources in the future.

"For the business enterprise, SD means adopting business strategies and activities that meet the needs of the enterprise and its stakeholders today while protecting, sustaining and enhancing the human and natural resources that will be needed in the future. This application of SD on the corporate level, which obviously builds on the Brundtland Report (WCED, 1987), is often referred to as Corporate Sustainability. While SD is commonly perceived as societal guiding model, which addresses a broad range of quality of life issues in the long term, CS is a corporate guiding model, addressing the short- and long-term economic, social and environmental performance of corporations. If one accepts this understanding of CS, the microeconomic framework of SD can also be read as a framework of CS" [47] (p. 274).

The works of D. Wheeler, B. Colbert, R.E. Freeman, T. Dyllick, W. Visser, K. Hockerts, M. Bergman, M. van Marrewijk, R. Likert, G. Goyder, P. Bansal, F. Székely, M. Knirsch, N. Finch, K.Y. Belousov, M. Beckmann, S. Brammer, I. Ivashkovskaya, L. Becchetti, and

many other researchers [28,49–62] are devoted to the further development of the ideas of corporate sustainability.

D. Wheeler, B. Colbert, and R.E. Freeman define a sustainable organization as the achievement by the organization of the highest level of organizational culture, in which there is synergy between the company, stakeholders, and value-oriented networks. The result of synergy is the maximization of the created values within the framework of the triune approach [49]. In our opinion, this approach is based on the right ideas, but it does not solve the problems of CS assessment and management.

In a later publication (2010), Elkington defines CS as follows: corporate sustainability is cross-conceptual and intersects with the categories of human development, corporate citizenship, environmental governance, ethics, stakeholder management, and social responsibility [50], i.e., it to a large extent coincides with CSR. In this regard, according to the author, CS cannot become the key goal of a company whose focus is economic development under the influence of extensive and intensive changes, including an increase in sales, profitability, capitalization, tangible and intangible assets, innovative projects, the rate of production modernization, etc.

"Corporate sustainability, then, is probably better understood not so much as the discipline by which companies ensure their own long -term survival—though that is clearly part of the equation—but as the field of thinking and practice by means of which companies and other business organisations work to extend the life expectancy of: ecosystems (and the natural resources they provide); societies (and the cultures and communities that underpin commercial activity); and economies (that provide the governance, financial and other market context for corporate competition and survival). By paying attention to such wider issues, it is often argued, companies are better placed to ensure that their own business models remain valid and adaptable" [50] (p. 115).

M. van Marrewijk and M. Were understood corporate sustainability as the company's activity aimed at integrating into the social and environmental problems of society. The key integration tool is the company's interaction with stakeholders [51]. The approach is based on the modern stakeholder framework, but it is unclear how it is possible to assess the level of integration or how the integration results in corporate sustainability.

"Corporate sustainability, and also CSR, refers to a company's activities—voluntary by definition—demonstrating the inclusion of social and environmental concerns in business operations and in interactions with stakeholders. This is the broad—some would say "vague"—definition of corporate sustainability" [51] (p. 95).

T. Dyllik and K. Hockerts also link the classical definition of SD adopted in the report "Our Common Future" and the triune approach with the interests of stakeholders. In their opinion, CS should be understood as meeting the interests and needs of current stakeholders without harming the interests and needs of future stakeholders [52]. In this formulation, a conflict between stakeholders is inevitable in terms of both the importance of their interests and the traditional conflict at the macro level, i.e., ensuring intergenerational equality. To some extent, the authors solve this problem when developing the concept of managing economic, natural, and social capital. It should be noted that Dyllik and Hockerts identified certain types of capital, but they did not sufficiently combine them with strategic management sufficiently, i.e., corporate sustainability is considered conceptually, without addressing the issues of CS assessment and management.

Under economic capital, Dyllik and Hockerts understood the income of the company that determines the framework of its current and future activities, linking the short and long-term goals of the company [52].

The authors understand natural capital as natural resources and ecosystem services. For many of the services provided by the natural environment, there are no known substitutes, or they are available only at prohibitive prices. Sustainable companies use only natural resources that are consumed at a rate below natural reproduction or at a rate below the development of substitutes. They do not produce emissions that accumulate in the environment at a rate that exceeds the ability of the natural system to absorb these

emissions. Finally, they do not engage in activities that degrade ecosystem services. In our opinion, this idea of environmentally sustainable companies is hypothetical. The consumption of resources at the rate of reproduction or at a rate below the level of development of substitutes, especially for mining companies, is not possible. Most natural resources are nonrenewable or there is no technology to produce substitutes.

This means that companies must limit their growth rate when using resources to the limits of current income and have to use resources intensively and improve efficiency. To some extent, this approach is related to the concept of "strong sustainability", which seems unrealistic in the current economic conditions but can be reached using circular economy models.

Social capital refers to social connections and networks that make it possible to effectively interact with stakeholders to achieve common goals. Social capital is manageable; the idea that companies should contribute to it appeared in the 1960s, at the dawn of the corporate social responsibility concept, in works by Lickert [53] and Goyder [54].

Thus, according to Dyllick and Hockerts, when transferring this idea to the business level, CS can be defined as meeting the needs of the company's direct and indirect stakeholders (shareholders, employees, customers, pressure groups, communities, etc.) without compromising its ability to meet the needs of future stakeholders. To achieve this goal, companies must maintain and grow their economic, social, and environmental capital base while actively promoting political sustainability.

"When transposing this idea to the business level, corporate sustainability can accordingly be defined as meeting the needs of a firm's direct and indirect stakeholders (such as shareholders, employees, clients, pressure groups, communities etc), without compromising its ability to meet the needs of future stakeholders as well. Towards this goal, firms have to maintain and grow their economic, social and environmental capital base while actively contributing to sustainability in the political domain" [52] (p. 131).

Bansal defined CS as a three-dimensional construct based on economic prosperity, social justice, and environmental integrity [55]. This approach is consistent with the SD concept of and the Brundtland definition, but it does not define the features of SD at the corporate level.

Székely and Knirsch defined CS as a way to maintain economic growth, shareholder value, prestige, and corporate reputation; to ensure customer relationships and the quality of products and services; to adopt and comply with ethical business practices; to create sustainable jobs; to create value for all stakeholders; and to meet the needs of the underprivileged [56]: "Companies embarking on a strategic approach to corporate sustainability expect their contributions to enhance business performance and to support the long-term interests of the company. The Global Compact Initiative 1 has identified a number of ways in which the efficient management of environmental, social and governance issues can contribute to creating shareholder value" [56] (p. 629). Being a conceptually correct approach, it also needs substantiation of the assessment tools and implementation practices.

Finch's work on CSD is based on the triune concept of SD and at the same time focuses on CSR [57], which is understood as part of sustainability. A direct definition is not given, but the term sustainability is used.

Despite the fact that many CS studies link CS and SD at the micro level, we believe it is necessary to distinguish between the concepts of CS and CSD.

The World Commission on Environment and Development has defined sustainability as economic development that meets the needs of the present generation without compromising the ability of future generations to meet their own needs. However, this macroeconomic definition does not provide guidance on how to implement this concept at the corporate level, and managers still wonder how to implement a strategy that promotes CS when there are many competing organizational constraints and numerous impediments to implementation [9].

This raises a question of whether it is possible for an organization to pursue SDGs without achieving CS. It is important to combine various approaches to understanding

the term "sustainability" in such a way as to ensure a broad understanding in the context of the overall goals of SD and the goals of the corporation.

An organization striving for CS must have a comprehensive and consistent sustainability strategy. This can be achieved through means such as company manuals or guidelines that clearly include and articulate the organization's sustainability strategy, operational guidelines that aim for sustainability, the organization's culture that produces sustainability principles, and efficient staffing [63].

Elkington rightly notes that the average life cycle of a company is too short to talk about its interest in achieving global SDGs and ensuring the ability of future generations to satisfy their needs. At the same time, corporate sustainability is vital to them. The life cycle of a company is understood as the period of its existence without changing ownership [44].

V. Anshin believes that CS is the microeconomic level of the macroeconomic concept of SD. Beckmann argues that CS issues, including the moral aspects of institutional legitimacy, should continue to be explored [64]. However, at the micro level, a "free rider effect" is possible, and companies are focused on their own economic goals, interests, and benefits.

Very important studies were conducted by Hart et al. They present the development of CS ideas in the context of SD over a 20-year period beginning in 1995. Hart's early work [65,66] articulated three phases of an active environmental strategy for companies, including pollution prevention, product stewardship and sustainability, and laying the foundation for a CS strategy.

First, while Hart outlined pollution prevention, product stewardship, and sustainable development as the three stages of proactive environmental strategy, the area of CSD strategy has since been separated into two distinct areas: clean technology and BoP.

Hart and Dowell [67] revisit (in 2011) Hart's (1997) natural-resource-based view of the firm (NRBV), summarize the progress made in testing elements of this theory, and revisit the NRBV in light of a number of important developments that have taken place in recent years in both resource-based literature and research on sustainable entrepreneurship. First, the authors consider how RBV can evolve in the context of dynamic capabilities, especially in rapidly changing environments. Second, they review the latest cleantech and business research at the base of the pyramid and suggest how NRBV can help inform researchers about the resources and opportunities needed to enter and succeed in these areas. Combining these perspectives improves the understanding of the role of environmental strategies and company success and links them to SD at the macro level.

Hart suggests that while pollution prevention and product stewardship can improve environmental efficiency, addressing global sustainability may require companies to actually reduce material and energy consumption in developed markets while creating markets in developing countries. Cleantech strategies are concerned with how firms create new competencies and position themselves for competitive advantage as their industries evolve. The reduction in material and energy consumption is driven by the pursuit of clean technologies that meet human needs without depleting the planet's resources.

Hart [66] made a clear distinction between "greening" strategies (pollution prevention and product management), which focus on the incremental improvement of today's products and processes, and "beyond greening" strategies (clean technology and sustainability), which focus on the future of technologies and markets. In recent years, the interest of scientists and practitioners in cleantech has grown along with entrepreneurship in renewable energy and other areas of cleantech. For the purposes of NRBV, a key cleantech issue is understanding what resources and firm capabilities can be associated with the effective commercialization of cleantech [68].

Valente also believes that the company should be sustainability-centric [69], moving towards a proactive sustainability strategy. Companies should strategically link social, economic, and ecological systems at different levels, bringing together stakeholders into a single network and system, and thereby creating a strategic competitive advantage that

competitors cannot identify and reproduce. The author's conviction that a new paradigm regarding CS is needed coincides with Hart's vision and can be considered progressive.

M. Bergman, Z. Bergman, and L. Berger also define CS as a business approach and strategy of a company, whose implementation achieves long-term social and environmental effects if the company's behavior is economically motivated for customers, employees, and shareholders. This approach makes it possible to identify the relationship between CS and the company's activities and, in the future, through the measurement of results, to evaluate both the level of CS and the quality of the implemented strategy (which is also an unresolved issue in modern literature on the topic): "Corporate sustainability refers to a systematic business approach and strategy that takes into consideration the long-term social and environmental impact of all economically motivated behaviors of a firm in the interest of consumers, employees, and owners or shareholders" [59] (p. 10).

*3.1. Basic Concepts of Sustainable Development at the Macro Level*

A long discussion of the three key SD concepts (environmental, economic, and triune) has led to the dominance of the triune concept [70].

From the point of view of the environmental approach, SD was considered by D. Meadows, G. Daly, R. Costanza, N.N. Moiseev, V. G. Gorshkov, K. Y. Kondratiev, K. S. Losev, V. I. Danilov-Danilyan, and other researchers [71]. According to the authoritative opinion of D. Meadows, SD in its original understanding is not scalable and can only be considered on a global scale (planet, region) [72]. Therefore, it should focus on environmental problems common to the whole world. This approach is focused on maintaining ecosystems at the level necessary to meet societal needs [70]. At the same time, the role of a person is ambiguously defined, on the one hand a part of the biosphere and on the other hand a user of resources with growing needs. Moreover, a person is a part of society and at the same time an economic agent. Under this logic, human activity is opposed to environmental development [73].

The key place in the environmental approach is taken by the development of natural capital [74], which is understood as the ability of the natural system to restore itself and adapt to changing conditions. The principle of environmental sustainability lies in the limited depreciation of natural capital, which takes the form of ecosystem services as well as the renewable and nonrenewable natural resources used in economic processes. Ecosystem services are the natural processes of the environment, which are analyzed from the perspective of "industrial metabolism" [75]. The social and economic components of SD, according to this approach, are in confrontation with environmental sustainability, while economic development is perceived as a threat and a cause of ecosystem degradation [76].

In 1987, H. Daly and R. Costanza in their work titled "Natural Capital and Sustainable Development" also proved that economic theories do not take into account natural capital. However, society must conserve natural capital and limit its consumption or limit growth in order to conserve resources, which according to the authors leads to strong sustainability. However, this condition can be interpreted in different ways: either as a complete ban on the reduction of natural capital or as an assumption of some reduction in it.

Contrasting views are reflected in the concept of weak sustainability [77], which recognizes the interchangeability of manmade and natural capital, which leads to a weakening of the factor of limited natural resources and an increase in the role of technological innovation. For example, in the book *Natural Capitalism: Creating the Next Industrial Revolution*, H. Lovins, A. Lovins, and P. Hawken criticize traditional industrial capitalism, arguing that there is a liquidation of capital, called income, that makes it necessary to recognize the interdependence between production, the use of human capital, and the maintenance of natural capital for the purpose of the long-term continuation of activities [78].

These environmental approach concepts focus on natural capital at the macro level. Given the importance of the global problem of the use of natural capital, mining companies are the subjects that exploit it. They receive and appropriate the economic results of the exploitation of natural capital after obtaining a license to extract minerals. Therefore, the issues of natural capital management should be considered at the micro level, and they determine corporate sustainability to a large extent.

Economic concepts include both neoclassical and institutional ideas. In highly developed neoclassical theory, natural resources and the environment take a secondary and subordinate position in relation to the economy. The ecosystem was considered a subsystem of the economy, the main functions of which are the unlimited extraction of resources and the free disposal of waste. The goal of development is continuous and exponential economic growth, which should increase innovation and efficiency; economic growth is not associated with a negative impact on the environment. The well-being of future generations is safe because the depletion of natural capital can be offset by investment in other forms of capital [79].

Neoclassical economists defined development as growth in social welfare. Welfare was measured through economic indicators, and it was found that the growth of the latter is not necessarily associated with an increase in the consumption of resources (materials and energy). Based on this, it was concluded that there are no contradictions between sustainability and development. The theoretical problems of this approach are related to the difficulties of combining an individual utility function into an aggregate one, making choice between generations, as well as the choice of indicators that measure social welfare and are not related to GDP. Neoclassical theory supports the concept of "weak sustainability", which assumes the interchangeability of natural, fixed, and human capital. The key economic mechanism is the price mechanism, which ensures effective management of externalities. Neoclassical models have greatly developed over a long period of evolution [77].

There are approaches that are characterized as alternative to neoclassical views [80]:

(1) Managerial approach. W. J. Baumol put forward the idea of a conflict of interest between company managers and shareholders. The former are interested in maximizing total revenue, while the latter are interested in maximizing profits. A conflict of interest may also relate to the achievement of CS and CSD. Shareholders are focused on achieving results in the long term, that is, in CSD. At the same time, it is important for managers to get the result as soon as possible in order to achieve KPIs, i.e., managers are focused on achieving CS.

(2) Behavioral theory of the firm. This theory assumes that the firm has many interests, the implementation of which is limited by factors external to the firm. At the moment, ESG factors and, as a result, standards developed based on these factors (for example, the SASB standards) can be identified as major limitations.

(3) The stakeholder theory. The firm is viewed as a network of its internal and external relationships with individuals and groups, and the emphasis is on the task of reconciling and managing their diverse and often conflicting interests. In our opinion, it is not advisable to take into account the interests of all stakeholder groups; this complicates the management process. To achieve CG and CSD, it is necessary to identify groups of the most influential stakeholders and take into account only the interests of those groups that have the strongest "weight".

The triune approach implies the achievement of SD in terms of a balanced combination of three aspects: environmental, social, and economic. To achieve SD in the environmental sphere, it is necessary to harmonize the interaction between man and the environment [81]. As a result, society will begin to maintain the viability of the biosphere and the integrity of natural ecosystems, implementing the strategy of civilization, which must be consistent with the strategy of nature [82]. The main goal of SD in the social sphere is to improve the quality of life: to achieve a free and equal society, maintain the stability of

social and cultural systems, increase employment rates, eliminate poverty and unemployment, and increase the role of the integration of man and civil societies into social processes. Economic sustainability refers to the creation of economic systems that rationally use the available limited resources. For corporate sustainability, this should encourage the use of resource- and energy-saving, low-waste, recycling technologies and the implementation of environmentally friendly projects [83].

Thus, a number of ideas of SD concepts at the macro level can be used for companies at the micro level to justify the effective use of natural capital, the introduction of BAT technologies, and the concept of recycling in mining companies, thus defining the relationship between SD at the macro and micro levels.

### 3.2. Concepts of Corporate Sustainable Development (CSD) and Circular Economy

Conceptually, circular economy can be viewed as an umbrella concept for resource and waste management based on the principles of several schools of thought. It combines various resource strategies and circular business models in an effort to maintain business viability, competitive advantages in the business sector, and the long-term sustainability of multilevel economic systems [19,84,85].

In the SD paradigm, CE conceptually supports the separation of economic growth from the use of scarce primary resources and the reduction of pressure on the environment associated with waste generated by consumption and production [19]. CE promotes the responsible and circular use of resources, potentially contributing to SD and economic growth through the creation of new businesses and jobs, saving materials, reducing price volatility, and increasing the reliability of supply while reducing the environmental burden [23,86]. There is still no consensus in the literature on the conceptualization of the principles and measures of circularity, [87] and moreover, there is no generally accepted concept of CE, which makes it difficult to establish an exact relationship between CE and sustainability [22,23].

Sustainability is subdivided into environmental, economic, social, and technical (technological) [88]. For modeling purposes, it is acceptable to assume that the effects in the environmental, economic, and social fields are caused by the technological cycles of materials, products, and services [23]. The "technological" feature of CE is that any waste streams are closed and returned as secondary resources to the production cycle in the production–consumption system. In addition, resource flows are being slowed down by companies using circular business models to extend the life and value of products, restore resources, and/or increase the use of products [89]. Closing and slowing down material flows leads to the saving of nonrenewable natural resources and, together with the energy transition, makes the economy more sustainable. Up to ten R-strategies are used to minimize waste and extend the life of products, materials and resources: Refuse, Rethink, Reduce, Reuse, Repair, Refurbish, Remanufacture, Repurpose, Recycle, Recovery [21,90]. There are also data on 45 CE strategies suitable for application in various parts of the value chain, and more than 100 cases of practical implementation of 35 of these strategies have been described [86].

The EU taxonomy recognizes CE, together with sustainability and climate neutrality, as one of the decisive factors for long-term competitiveness at the national level (EU countries). The transition to CE is classified by the EU Taxonomy as one of the six environmental goals (together with climate change mitigation and adaptation, the protection of water and marine resources, pollution prevention and control, and the restoration of biodiversity and ecosystems) supported by sustainable financing [91].

In the literature, sustainability and CE are seen as interrelated and interdependent disciplines [92]. At least eight different types of relationship between sustainability and CE have been identified, and an innovative aspect of SD based on CE components has been highlighted [93]. There are studies arguing that business sustainability considerations from an ESG perspective can keep operating models circular. The concept is also

being developed that ESG reporting serves as a tool through which business operations can drive circularity and remove the constraints of the linear economy in practice [94].

An important aspect of circularity for supporting sustainability and SD at the micro level is its qualitative and quantitative certainty. More than two dozen indicators and models for assessing CE on a microscale have been proposed in the literature [23]. Most of the indicators focus on strategies related to the conservation of materials through recycling, including economic value creation [23]. Some indicators are complex, for example, the sustainable circular index (SCI), which is determined based on the analysis of sustainability reports (TBL, GRI) [95]. To complement CS efforts, the World Business Council for Sustainable Development (WBCSD) launched the V2.0 Circularity Metrics in 2021, allowing companies to periodically assess their circularity performance, associated linearity risks, and recycling opportunities.

The weak point of the CE approach in the context of sustainability remains the lack of "authoritative guidelines" on key CE issues at the organizational level [19]. In 2017, the British Standards Institute published the world's first standard in the field of CE: BS 8001:2017—Framework for the implementation of the principles of the circular economy in organizations. The standard defines terminology, a set of general principles, and a flexible management structure for the implementation of circularity strategies, as well as other issues, and it provides for the full responsibility of organizations for the selection of appropriate CE indicators [19]. The standard defines that when implementing the principles of CE, the key goal of an organization is to create long-term business value through the sustainable management of the resources of its products and services [96]. However, despite the presence in the standard of procedures for linking the principles of CE with the sustainability of the organization, it does not provide an explanation of the relationships between CE and sustainability or SD [97]. Clarifying the links between CE, sustainability, social risk, and ethical responsibility will require a broader discussion.

Thus, the concepts of CE, sustainability, and SD at the micro level do not contradict each other but are consistent in their basic interpretations. For example, Hart and Dowell's strategic approach is to support the sustainable development of the company indefinitely into the future by minimizing environmental damage and reusing products, which is consistent with the principle of the responsible and circular use of the organization's resources. CS in the interpretation of M. Bergman, Z. Bergman, and L. Berger is strategically aimed at taking into account the long-term social and environmental consequences of the company's economically motivated actions in the interests of its key stakeholders, which also does not contradict the concept, principles, or strategies of CE. An approach with a multi-aspect definition and assessment of CS involves developing a set of characteristics for assessing CS, one of which can be a general indicator of the circularity of the company's operating models. The inclusion of the circularity indicator in sustainability reporting in the context of ESG, in our opinion, can have a stimulating effect on the introduction of circular business models for companies in various industries. However, the justification, selection, and procedure for introducing industrial indicators of circularity into standard reporting, for example SASB, requires separate comprehensive studies.

### 3.3. Concepts of Corporate Sustainable Development and Corporate Social Responsibility

The concept of socially responsible business began to take shape in the 1960s and the 1970s. According to this concept, business should not only think about profits and pay taxes but also share responsibility for social injustice, economic inequality, and environmental problems with society. At the same time, the measure of responsibility remains a debatable issue, including many approaches to its solution, the main of which are the theories of corporate egoism, corporate altruism, and reasonable (enlightened) egoism.

According to the theory of corporate egoism, the only responsibility of a business is to use resources to increase profits for shareholders. In contrast, the theory of corporate altruism presupposes the obligatory participation of companies in improving the quality

of life of society. The theory of reasonable egoism is the most common compromise between them, believing that despite the reduction in the operating profit of the company as a result of the implementation of social programs, in the long term, a socioeconomic environment favorable for SD is created. Support for this concept is given by modern CSR standards (AA1000S), which provide for interaction with stakeholders and the public assessment of the level of socially approved business activities.

CSR is especially important for mining companies as they are characterized by difficult working conditions, city forming, budget forming, and social significance, which leads to an increase in the requirements for their social responsibility and environmental policy. The importance of the mineral resources sector in Russia imposes on mining companies great expectations from stakeholders to participate in society, as well as mandatory requirements for the safety and improvement of working conditions, participation in infrastructure projects, and the construction of social facilities [98]. In other countries, it is increasingly demanded from mining companies that they implement environmental programs.

The main attention to CSR issues as the main factor of CS was given by Elkington, Steuer, and Dyllick and Hockerts [44,47,52]. From the standpoint of CSR, SD is understood as the state of the company, in which the needs of stakeholders are simultaneously satisfied and the company's profit is maximized [99]. This is a very modern view, and it raises the question of choosing an indicator that satisfies the key stakeholders.

Taking into account some contradictions and opposition between stakeholder interests and profit maximization, studies on the issue can be divided into two groups focusing on:

- the interests of owners in improving the quantitative economic indicators of SD (financial results and growth in the value of companies);
- the interests of a wider range of stakeholders, the search for competitive advantages, and the connection between CSR and the strategic development of companies.

Among the first group of works, most researchers tried to show the positive impact of CSR on financial results. The study, published in *Portfolio Management* [100], showed the positive impact of ESG and CSR requirements on financial performance and capital valuation. The authors of the study explain this both through systemic risks (lower cost of raising capital and larger valuation multiples) and through nonsystemic risks [101] (higher profitability, lower vulnerability to residual risk). Brammer's work reveals the impact of CSR on the profitability of company shares in the short and long terms using the example of 100 socially responsible American companies, which according to the author leads the company to financial stability [60]. In the period from the 1980s to the 2000s, about 50 studies were conducted to identify the relationship between a company's financial performance and CSR. The results of most empirical studies have shown that it is impossible to identify an unambiguous relationship due to the multiple influences of CSR on the objects of study. If any relationship between financial performance and CSR was identified, then it was determined accidentally. Despite the fact that most studies at the end of the period under review revealed the presence of a probable correlation, the relationship between CSR and financial performance has not been convincingly proven [102].

The second group of works presents studies in the field of CSR impacts on competitive advantages, which should be taken into account in strategic management. This connection is considered by Jr. Werther and D. Chandler [103], D. Melé [104], L. Becchetti [61], O.A. Romanova [105], I.V. Ivashkovskaya [62], B.A. Shakhnazaryan [106], and many other researchers.

Becchetti's studies developed ideas that the implementation of CSR in the company's management system reorients the company's goals from the interests of shareholders with the maximization of their welfare to the interests of stakeholders. As a result of CSR, the author saw a decrease in the volatility of profitability, an increase in sales with a decrease in the profitability of shares, and an increase in corporate stability. At the same time, CSR

has both positive and negative effects: Among the positive ones, one can distinguish an increase in employee motivation, an increase in productivity, and an increase in sales, while rising costs are among the negative ones [61]. According to Ivashkovskaya, value creation for stakeholders (STV) must be linked to the principle of economic profit. As applied to nonfinancial stakeholders, this criterion is based on the principle of the effective realization of their interests in the form of benefits received compared with opportunity costs. As carriers of material resources and intellectual and social capital, they claim to receive remuneration from the company that compensates for their own opportunity costs that arise from their market opportunities. Nonfinancial stakeholders, like financial ones, expect to receive economic profits not lower than zero. When there is a situation of value destruction, nonfinancial stakeholders, like financial ones, lose the motivation for continuing to participate in communications with the company or may withdraw from them. The decision to exit depends on the exit costs, which are specific to a particular category of stakeholders [62], and the costs are higher with more-specific relationships, which is typical for specific mineral assets in mining companies.

The main goal of implementing CSR programs, according to many researchers, is to increase the competitiveness of a company that is ready for modern challenges. One can partially agree with this approach because in modern conditions, CSR contributes to the accumulation of reputational capital, stimulating the growth of trust in and loyalty to the company through interaction with the company's stakeholders and strengthening its stability due to the created competitive advantage [107]. At the same time, the internal and external environment of the company has a multifactorial influence. We can talk about creating a competitive advantage only for those companies whose stakeholders are focused on a commitment to SD principles. It is difficult to conclude to what extent management decisions depend on the opinions and needs of stakeholders [108].

The general ideas of CSD concepts in the context of CSR are:

- Companies must be actively integrated into all social processes, and must also participate in their management;
- The development of the social sphere leads to an increase in the economic stability of companies;
- Companies implementing CSR programs acquire some competitive advantages.

According to the SASB advisory group, companies in the resource industries can apply various community engagement strategies in their operations to manage the risks and opportunities associated with community rights and interests. Companies are beginning to adopt a "shared values" approach to provide a key social and economic benefit to society, allowing the companies to operate at a profit [109,110].

Professor B. Eckles (BCG) believes that CSR is characterized by dynamic materiality as follows: increasing evidence of why a social problem is significant leads to an escalation of stakeholder activity, which forces companies to address this problem so that they do not lose customers and truly engaged employees. Investors who understand the implications for profitability are also becoming more proactive in their engagement with companies on this issue. For example, COVID-19 went from an almost nonexistent ESG issue in mid-January 2021 to over 60% of the total amount of information on SASB issues three months later [111].

In the well-known article titled "Creating Shared Value", M. Porter and M. Kramer came to the conclusion that companies can go beyond CSR and gain a competitive advantage by including social and environmental considerations in their strategies. In their opinion, viewing social problems as business opportunities is the most important new dimension of corporate strategy and the most effective path to social progress. The central premise of creating shared value is that a company's competitiveness and the health of its surrounding communities are interdependent. Recognizing and exploiting these links between social and economic progress can lead to global growth and change the understanding of capitalism [112].

*3.4. Studies Focusing on the Impact of the Distinguishing Features of the Mining Industry on CS in the Mining Sector*

Researchers on corporate sustainability in the mineral resources sector highlight the lack of industry-specific scientific publications. At the same time, industry specifics are taken into account in ESG. The studies on the topic can be divided into two groups:

- Studies that do not take into account the significant difference between the mining industry and other industries, so the CS of mining companies is considered from the point of view of a general approach to CS;
- Studies taking into account that extractive industries have their distinguishing features that have significant impacts on the environment, economy, and society. For example, for oil and gas companies, special indicators are required taking into account the industry's features [27] since universal indicators for processing industries are not enough.

It is difficult to agree with the first group of studies because mining companies are undoubtedly a specific object of study. The second approach is justified, but unfortunately is in the initial stage of development. Therefore, it is required to substantiate the selected indicators for mining companies under the current SD conditions [113].

The specificity of the mining industry is considered only in the context of SD assessment indicators for mining companies, which requires further research [114]. Particular attention is paid to environmental indicators (the international report titled *Mapping Mining to the Sustainable Development Goals: An Atlas* [115]).

SASB is the world's first collection of SD standards that takes into account industry specifics, which makes it possible to analyze opportunities for long-term sustainable value creation by identifying significant factors influencing each industry sector [29]. SASB standards have been developed for eleven industry sectors, including industries with the greatest impacts on the environment and socioeconomic development: mining, metallurgy, oil and gas production.

Orientation towards the development of industrial criteria for evaluating Russian companies is reflected in the Concept for the Development of Public Non-Financial Reporting [116]. Baseline indicators were identified in the economic, environmental, and social spheres. At the second stage, a systematic assessment of companies was carried out based on public nonfinancial reporting, and a rating of companies in the SD field was compiled. At the third stage (2021–2022), it is planned to develop additional criteria taking into account industry specifics; at the final stage (from 2023), the list of companies obliged to publish sustainability reports should be clarified.

*3.5. The Mining Industry, Sustainability, and Circular Economy*

Although it has a high, albeit specific, potential for circularity, which can mainly be seen in mining and mineral processing waste including used water, the mining industry is not part of CE models [117]. However, the high levels of energy consumption as well as the major environmental and social impacts produced by mining companies make relevant such issues as sustainable and environmentally friendly mining practices, decarbonization, energy efficiency, and related risks relevant to the industry.

The first major study of the mining sector in the SD context was carried out by the IIED [118] with the support of WBCSD [119], and it resulted in the Minerals Sector and Sustainable Development framework (MMSD, the Mining, Minerals, and Sustainable Development Project, 2002), which includes economic, environmental, social, and governance aspects of sustainability. An integrated approach to the use of mineral resources began to be considered as a separate focus area for the mining industry to become sustainable. This approach involves recovering, reusing, and recycling waste as CE strategies. The ICMM report (2016) analyzes the role of mining and metallurgy in CE and the possible

consequences of the transition to a CE model for the mining sector [120]. The path of following the Green New Deal and the CE principles may hide both threats and new opportunities for the mining sector [117].

The discussions at the World Circular Economy Forum (2020) showed that there are fewer problems than opportunities associated with the transition to CE principles in the mining sector, and the case studies presented at the forum by Natural Resources Canada, Agnico Eagle, and Anglo American put the CE theory into practice [121]. The application of circular approaches in the mining sector remains insufficient in the context of sustainable mining and circular business models that need to be adapted to specific mining conditions and inter-industry interactions. The application of circular approaches in the mining industry remains poorly understood in the context of sustainable mining and circular business models, which must be adapted to specific mining conditions and interactions between different sectors of the economy. A literature review showed that the green and climate-smart mining (GCSM) model [122], which includes the circular processing of mineral resources and establishes a link between mining and sustainable development research, is promising [123]. A system of indicators for assessing open-pit mines using the GCSM model was developed and tested, and it was concluded that the integration of a circular approach, green supply chains of minerals, and a green procurement policy into the GCSM model will increase the environmental efficiency of mining and the sustainability of mining companies [124].

The energy transition as part of the green transformation has dramatically increased the demand for critical minerals and metals to build a low-carbon energy infrastructure, which creates the problem of supply risks [125] and requires the rethinking of value chains and the transition of mining companies to more sustainable circular business models [126]. To identify, assess, and manage the potential risks associated with supplying minerals critical for the green energy concept, a toolkit has been proposed [124] that is a factor in both corporate sustainability and the sustainable development of the low-carbon energy sector.

In the process of green transformation and deepening ESG reporting, mining companies are currently facing increasing pressure from stakeholders and suppliers, as well as growing challenges related to the environmental and social impacts of their activities [126].

According to an EY report, in 2021, ESG risks came first ahead of the risks associated with decarbonization and obtaining licenses for the first time in mining industry risk assessment. As ESG factors become more of a priority for investors, shareholders, and wider stakeholder groups, mining companies are looking to integrate ESG into corporate strategies, decision-making processes, and reporting to stakeholders [127]. Mining companies that can demonstrate their contribution to sustainable development will gain a competitive advantage [128]. Analysts show that companies with higher ESG rankings have an average total shareholder return that is 10% above market returns   and have better access to capital .

In risk rankings [128] the business models used were the most important risk factor. The EY report outlines six models that can help mining companies add value in a volatile environment, including the waste minimization and emission reduction circular business model [128]. It includes the application of CE principles in mines from the mining site to the recycling processes in order to minimize waste. The operation of mining facilities during the period while mining operations can be conducted at acceptable environmental costs while minimizing the loss of nonrenewable resources can be considered the contribution of the mining industry to achieving CE goals [24]. Environmental efficiency and sustainability have been identified as key characteristics of sustainable mining, where mining optimization and minimizing the amount of valuable components in the waste will help to solve problems such as deteriorating ore quality, falling economic efficiency, and increasing production volumes [129]. The mining industry will need to gradually move towards "closing-the-loop" strategies that will significantly reduce waste [130].

Mining and mineral processing waste volumes can be decreased based on compliance with the 3R principle using the experience of China, as well as through the introduction of breakthrough innovations [131]. In China, a mining circular economy used to be interpreted mainly in the environmental aspect as an economic system that is based on the highly efficient exploitation of mineral deposits and the integrated use of mineral resources [132]. The CE model of the mining industry is a closed-loop material flow (mineral resources—mineral products—renewable mineral resources) in the process of exploration, mining, processing, smelting, advanced processing, consumption, and other processes, on which the flow of energy and the flow of information are superimposed [133].

In 2021, China adopted a new development plan to make CE a national priority in the 14th Five-Year Plan 2021–2025, which will strengthen the "dual circulation" paradigm for the development of domestic and foreign markets. Efforts will be made to promote the green transformation of the economy and society based on green, low-carbon, and circular development [134].

To promote the concept and elements of CE in mining activities, a systematic view of this problem supported by theory is needed. Canadian scientists have identified six key CE principles related to mining: inventory optimization, i.e., increasing the value of materials; environmental efficiency; eliminating the concept of waste by expanding the value of resources; extended producer responsibility; the circular design of products and processes; and creating social value. The practical implementation of these principles in the transition from a linear economy to a circular economy in the mining and metallurgical sector involves three main actions: the development of circular operations; the introduction of new products and services based on the principles of circular production; cooperation with customers and building an ecosystem of circular manufacturing partners [128]. To increase the corporate sustainability of mining companies, it is necessary to integrate circular business models into corporate strategies. These models need to focus on reducing mining waste, creating closed-loop supply chains of secondary resources, and decarbonizing the energy supply to the main and auxiliary processes.

## 4. Results and Discussion

### 4.1. The Essence of Corporate Sustainability (CS)

Taking into account the results of the literature analysis and the classification of approaches to CS, we concluded that there is still no generally recognized study that defines CS as a concept that combines the concepts of SD, CSR, and stakeholders. The variety of interpretations of CS is due to differences in the understanding of the specific features, management goals, and key elements of CS. This signals that there is no universally accepted system of views on what SD is at the micro level. Furthermore, the theoretical foundations of the SD concept for various levels are still being developed, and there are yet unresolved problems of systematizing and integrating the existing approaches to the interpretation of CS.

In our opinion, the analysis of the key views in the field of CS allows us to conclude that there is no mainstream framework. The inconsistency of definitions and interpretations has so far prevented the development of high-quality and comprehensive tools for measuring the results and managing CS [135] in companies, and neither has it provided for taking into account what sector the company operates in. The lack of a universal approach to defining CS and the relationships of CS with sustainable development, CSR, and ESG factors in strategy development are the foundation of the CS methodology and the development of assessment and management tools.

In our opinion, the studies of the CS concept that were analyzed can be divided into four groups (Figure 3).

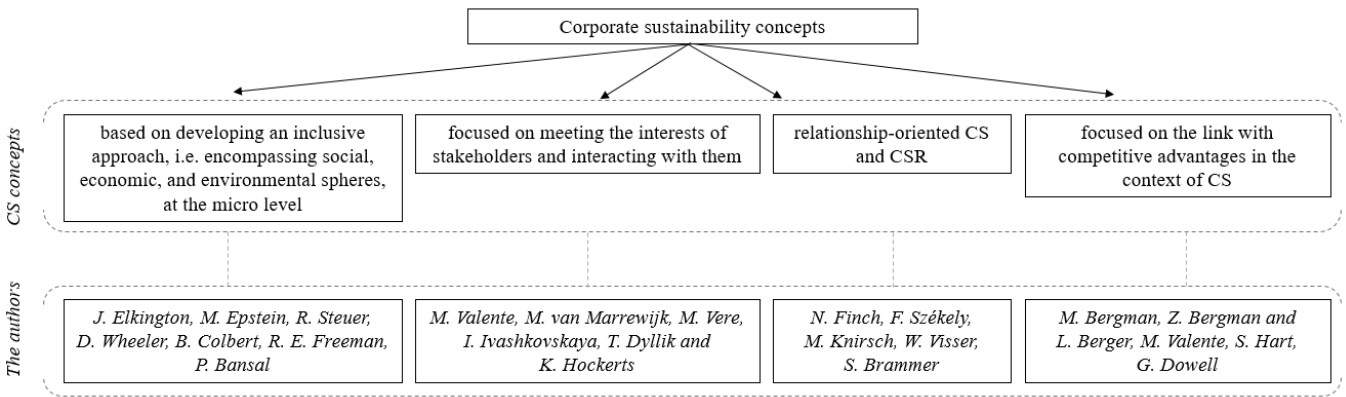

**Figure 3.** CS concepts (developed by the authors).

Our classification can be compared with a classification of scientific publications and approaches to determining CS by M. Bergman, Z. Bergman, and L. Berger (2017) [58]:

(1) Scientific publications that combine CSR and CS:
- CS and CSR are equivalent;
- CS and CSR are different;
- CS is achieved with the help of CSR tools.

(2) Approaches focusing on one aspect of determining CS based on:
- The observance of moral standards;
- The strategic management of the company.

(3) Approaches with multidimensional determination and assessment of CS:
- CS is defined through many characteristics (economic growth, product quality, business reputation, organizational structure, relationships with stakeholders, environmental protection, and many others at the same time);
- CS is considered within the framework of the triune SD approach;
- CS is the driver of a company's economic growth;
- CS combines various aspects of the company's activities and can be measured using specialized stock indices that take into account sustainability (for example, Dow Jones Sustainability Index, Shanghai Stock Exchange Sustainable Development Industry Index).

In our opinion, the presented classification has a number of shortcomings. First, the authors analyzed publications in various scientific disciplines, including ethics, leadership, organizational behavior, economics, business, and environmental management, so the definitions reflect CS from the point of view of one science. Second, the analysis in this work is limited to ten review articles, so all possible approaches are not fully reflected. As can be seen from the analysis, the approaches used by researchers to defining and classifying corporate sustainability correlate with the classification of approaches that we present (Figure 3). However, researchers do not factor in the distinguishing features of the industry. In our opinion, the following are the most important conclusions from the analysis. The common feature of all views on CS is the understanding that CS concepts are aimed at the long-term goals of the company (Hart and Dowell) and take into account the interests of stakeholders (Ivashkovskaya, Becchetti, etc.). Even though the long-term orientation of CS is an important feature, it is only declared so in most works without providing any rationale for periods of time and the relationship between short-term and long-term goals. In today's conditions of SD, reliance on the stakeholder theory and taking into account the interests of stakeholders is an important aspect of CS. The problem here is the

conflict between the interests of many stakeholders. A classic example is the tension between the short-term interests of managers to maximize bonuses and the long-term interests of shareholders to maximize shareholder value.

### 4.2. Correlation between the Concepts of CSD and CS

Research showed that Russian researchers often use the term "corporate sustainability" as a synonym for sustainable development [136]. The authors analyzed a similar study with the results of data analysis conducted by I. Montiel and J. Delgado-Ceballos, which also showed that the search for articles/abstracts from 1995 to 2013 with the keyword 'sustainab\*' did not lead to an unambiguous interpretation of the concepts of SD and CS [137]. Most of the definitions analyzed in the article traditionally link CS to the achievement of sustainability in three areas: environment, society, and economy.

Epstein's model of the relationship between sustainability factors and financial performance of a company differentiates between CS and CSD [46], which in our opinion is completely valid. However, for the mining industry, Epstein's model must be supplemented. As sustainability indicators, the authors suggest assessing mining companies and metal producers using indicators from SASB standards, as well as supplementing stakeholder groups (Figure 4).

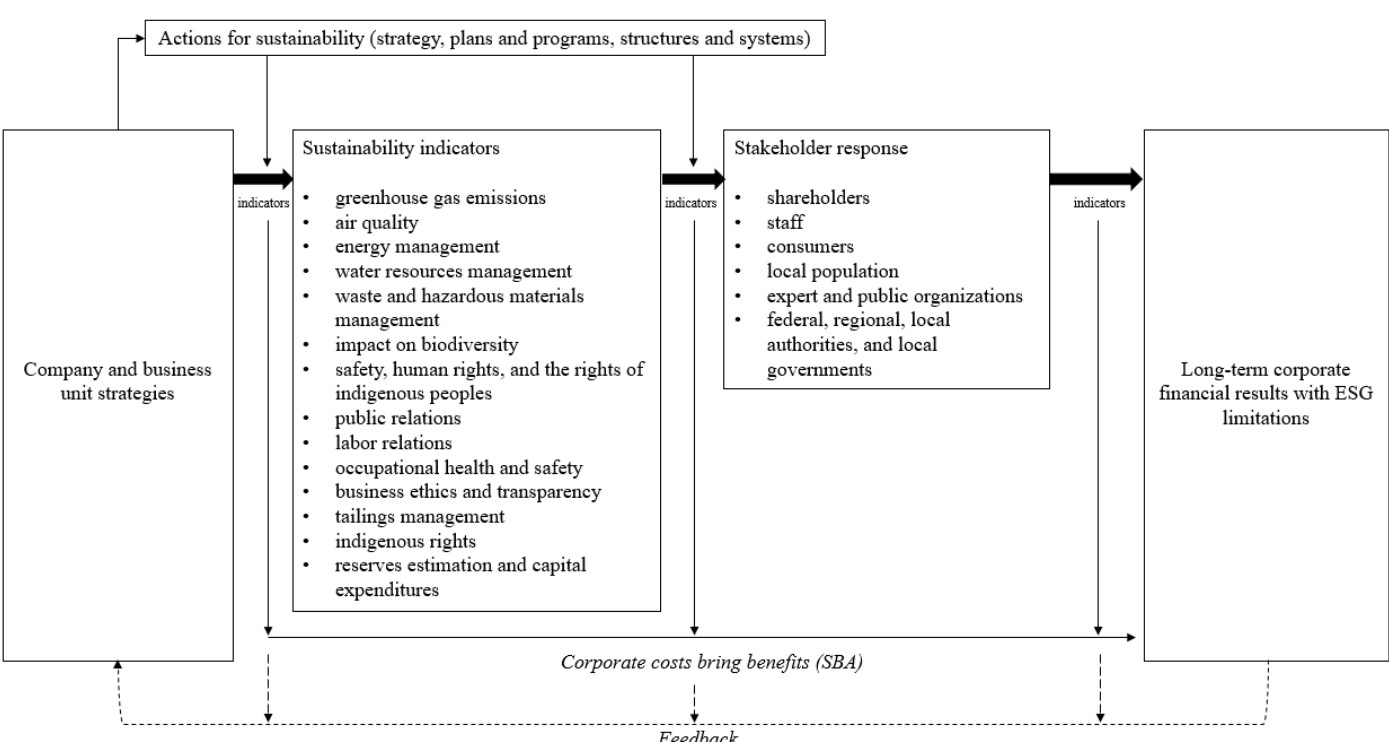

**Figure 4.** Sustainability factors and financial indicators (developed by the authors).

In our opinion, there are the following correlations between the concepts of SD, CS, and CSD:

- Macro level: SD;
- Micro level: CS or CSD.

At the macro level, the assessment of SD indicators is applied at the national level based on the methods provided by international organizations. The level of SD increase with the involvement of the government and the use of government regulation methods.

At the micro level, indicators are evaluated at the level of individual companies by industry taking into account legal regulations and targeted programs, as well as incentives

and preferences. The level of CS increases through strategic management in companies taking into account the integration of SD principles into strategies.

The concept of CS does not imply an exclusive focus on the economic efficiency and economic sustainability of the company. The complexity of CS (the development of the company in three aspects: environmental, economic, and social) implies a transition from business models that harm the environment to cleaner, circular models. However, such a transition may run counter to the economic efficiency of the company in the short term, i.e., there is a conflict between the SDGs and the economic interests of companies. The trend of making companies more responsible to society and their greater involvement in social development leads to an increase in the costs of implementing CS projects [136], which can lead to long-term effects and make the company better understand its impact on the development of society. For mining companies, the relationship between indicators and results in CE models is more complex. It has not been studied thoroughly enough and requires additional research.

As shown by the analysis performed by the current authors, some researchers believe that CS has secondary goals for the development of social, environmental, and other areas and also promotes wider interaction with stakeholders. The key CS tools are the normative method as well as descriptive methods. This topic also requires further research.

The development of international theory and practice in the field of CS in 2021 produced the following results: reorganization of regulatory bodies (IIRC and SASB were reorganized into the Value Reporting Foundation); updating and developing tools for assessing, managing, and disclosing the value of companies based on integrated thinking; integrated reporting; and developing SASB's sustainability reporting standards. With these key tools, organizations and investors around the world will gain a single, consistent, and shared understanding of what constitutes company value, which is created, destroyed, or maintained unchanged over time [138] under the influence of CS factors and associated risks.

In the academic literature, the issue of distinguishing between short-term and long-term CS has been little considered. According to Steuer's research, long-term CS allows for maintaining and improving the competitiveness and performance of the company, while short-term CS is focused on short-term indicators [47]. In the studies by Steuer [47] and Makarchenko, CS is understood as a tool for using opportunities to create shared value while continuously improving short-term efficiency and long-term growth [139]. Thus, short-, medium-, and long-term types of CS are considered in conjunction. The relationship between the processes of achieving CS was studied by N. Finch, who proposed a management pyramid.

In our opinion, this diagram logically combines the ideas of SD, ESG, and CS. However, the sequence of CS components seems questionable as it does not seem reasonable to determine which factor is more important in relation to another. The pyramid proposed by Finch can be supplemented by active interaction with stakeholders, the development of strategic management and corporate culture, the choice of indicators, and the development of a methodology for assessing CS to analyze the company's changes in development. In addition, strategies for achieving CS should be substantiated for both the long and short terms taking into account the distinguishing features of the industry and modern opportunities for the development of a circular economy. Thus, for the mining sector, the concepts of CS and CSD can be developed and presented as follows in Figure 5.

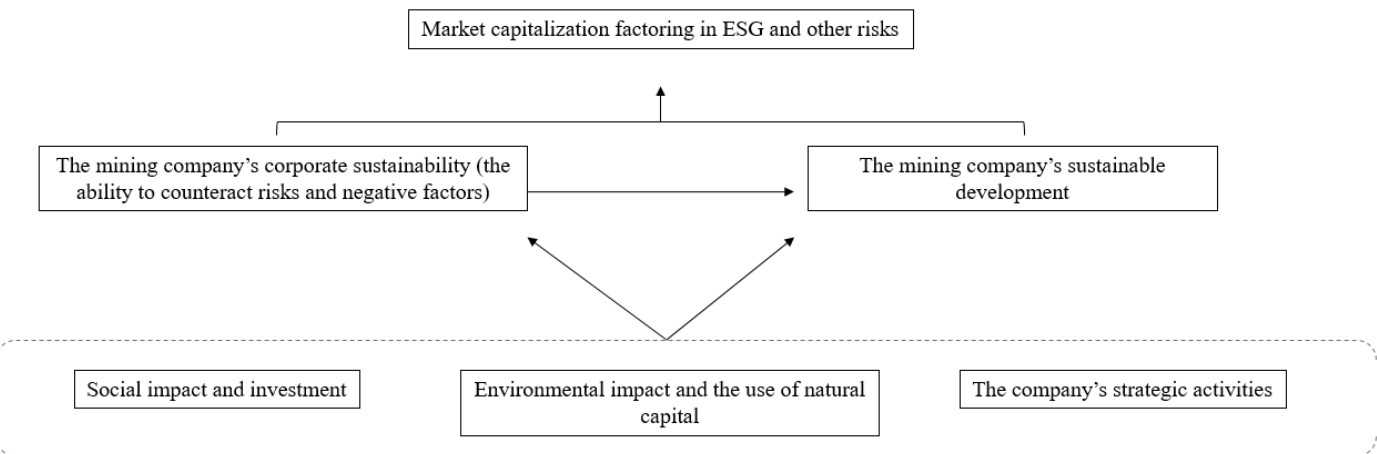

**Figure 5.** The relationships between CS and CSD in the mining sector.

*4.3. A Refined Interpretation of CS and the Distinguishing Features of Mining Companies*

The definitions of CS analyzed above do not take into account the nature of the mining industry.

As has been shown, mining companies have a major impact on the environmental, economic, and social development of mining regions and countries with resource-based economies (RBEs), which should be taken into account when determining CS.

First, the mining industry is associated with high environmental risks. Ecosystem deterioration (increasing emissions; water, air, and soil pollution) and natural resource depletion make companies introduce resource saving and switch to environmentally friendly technologies in the CE framework. Examples of how the mining industry affects the environment include the following [34]:

1.  Soil disturbance: alienation of agricultural land, changes in soil hydraulic properties, excavations, changes in the structural composition of the soil, water contamination by suspended particles [140].
2.  Mass movement and other geological phenomena: rock mass displacement, collapses, landslides, cave-ins, rock bursts, mudflows, erosion, deflation, suffusion, mud rushes, manmade earthquakes, seismic activity [141,142].
3.  Deformation of the surface, inter-aquifer mixing, pollution of groundwater and water bodies, decrease in the groundwater level and pressure.
4.  Changes in the composition and properties of air, fog occurrence.
5.  Soil and water acidification, salinization, pollution, changes in chemical composition.
6.  Decrease in the biological productivity of crops, degradation of flora and fauna. The most dangerous agents are heavy metals that produce toxic components. A major potential threat to the environment is metal deposits, which usually contain different elements whose concentrations are monitored in residential areas. Not all of them are extracted; they are usually stored in dumps, poisoning the environment with chemical agents.
7.  Ground disturbance: new forms of terrain emerge, terrain formation processes become activated [143,144].

Due to the influence of atmospheric and cosmic factors, rock dumps become chemically transformed and eroded. One hectare of rock dumps results in 200 to 500 tons of rock mass being carried over annually. Tailing impoundments become dust sources and infiltration zones through which dissolved minerals leech into the underlying rocks and groundwater. The growth in the volumes of explosion products generated by blasting operations in the mining sector causes the redistribution of stresses in the earth's crust and creates water and gas flows. Gas emissions affect the radiation balance and cause fogs, clouds, heavy rains, and catastrophic rainfalls.

The areas affected by mining and metal production are much bigger (by a factor of 2 to 10) than the areas of mining claims. Air becomes polluted within a radius of over 10–15 km from the source, and the radius of impact on the groundwater system is 20 km.

As pollution levels in the atmosphere, hydrosphere, and lithosphere depend on each other and there is an upward trend in the number and volume of pollutants, manmade ecological disasters are bound to happen [143].

The exploration of deep or hard-to-reach deposits is associated with the occurrence of manmade disasters and a decrease in the safety of underground mining, including a greater global risk and a greater local risk for workers operating mining machines or vehicles [142,145].

Manmade disasters have different forms: emergencies caused by drilling and blasting; manmade earthquakes; sinkhole formation; mine flooding. Hard rock blasting and manmade earthquakes cause vibrations that affect underground and surface structures as well as the environment [146].

Other major problems and challenges for mining regions are connected with excessive mineral extraction, the handling of sulfide-bearing wastes, and groundwater system control [146].

Mine waste generated in the course of mineral and metallurgical processing is stored on the surface and occupies vast territories. It has not only horizontal but also vertical expansion. Mine waste stored on the surface has a major impact on the ecosphere [146].

Manmade impacts on the environment in the mining regions have reached a level that exceeds natural recovery rates.

Among different industries, mining has a strong anthropogenic impact on the environment, which makes environmental management systems (EMSs) important. Effective environmental management aims to improve the environmental performance of industrial companies as an environmental aspect of corporate sustainability. Studies show that introducing an EMS [147] can make a company more competitive, which is critical for the mining industry. A higher level of competitiveness is associated with better working conditions and stimulates employment, which constitutes the social aspect of corporate sustainability.

It is important to note that the impacts that industrial producers, the service sector, and consumers have on the environment are constantly growing, making it necessary to take measures aimed at decarbonization and waste recycling in production and consumption [148].

Mine waste management issues have been relevant in the context of the circular economy for many years. For example, following the adoption of the Extractive Waste Directive (EWD) (2006/21/EC) in the EU, businesses have been submitting Extractive Waste Management Plans (EWMPs) along with their applications for environmental permits since 1 May 2008. A wealth of experience has accumulated, and the European Commission has launched studies to identify best EWMP practices. The first result of the studies was the development of a guidance document on best practices in the Extractive Waste Management Plans (Circular Economy Action; 2019). This document is based on the circular economy concept and focuses on two circular strategies: the prevention or reduction of extractive waste production and its harmfulness and the recovery or restoration of the value of extractive waste by means of recycling, reusing, or reclaiming such waste [149].

Second, the key resource that mining companies have is natural capital, which is part of the public natural resources involved in the economic turnover and an economic asset of the mining company. Therefore, the company creates economic effects, added value, and company value by using this resource. The contradiction lies in the interests of the company and society as an increase in the consumption of mineral resources can hinder the development of future generations, reducing sustainability at the macro level. Thus, the effective management of natural capital should cover the balanced interests of the company and society, contributing to the improvement of CS, CSD, and SD. At the same time, an unresolved problem is the lack of sound methods for a fair assessment of natural

capital at both the macro and micro levels, which is especially important for mining companies. For the macro level, such an assessment is based on determining the socioeconomic efficiency of the implementation of projects for the development of mineral deposits. For individual mining companies, it is based on commercial efficiency.

Third, mining companies differ from companies in other industries in the need for government regulation. On the one hand, the transition to sustainable development in Russia, the ratification of international climate agreements, the signing of international documents in the field of sustainable development, and the understanding of the possibilities of the CE stimulate the implementation of projects aimed at improving CS. On the other hand, it is important for RBEs to ensure the economic efficiency of extractive companies and the maximization of added value.

Fourth, mining companies are mainly large business structures that have the opportunity to implement CS and CSD projects. Therefore, such companies can be considered the major drivers for achieving SD at the macro level, especially in countries with RBEs.

Fifth, most of the social and environmental consequences of the activities performed by mining companies are long-term and are also resolved over a long period of time. Therefore, the CS management system in mining companies should be multi-level, ensuring the links between short-term and long-term goals, CS and CSD. Thus, the implementation of CS actions in the CSD strategy at mining companies requires setting and agreeing on appropriate goals.

Finally, a challenge for mining companies is the creation and transformation of value-added chains (VACs) and value-oriented networks. On the one hand, it is extremely difficult for mining companies to control the full chain of multistage supply from the purchase of materials and services to the production and delivery of products to consumers and end users. On the other hand, it is difficult to choose contractors providing specialized services in the mining industry due to their limited number.

The following distinguishing features of mining companies should be taken into account in the development of strategies and implementation of programs to improve CS and CSD:

- Mining companies actively support and increase mineral assets, which is necessary for long-term goals (CSD);
- They efficiently and rationally use natural capital, including mineral assets, as well as soil, land, water, and forest resources, which is necessary for CS and CSD;
- They develop and implement CSR strategies based on a combination of balanced interests of stakeholders, which is manifested in CS;
- They incorporate elements of circular business models in their strategies, including the use of waste-free and low-waste technologies, recycling, and the multipurpose use of mineral resources and subsoil, which will influence CS and CSD.

Taking into account the nature of the activities performed by mining companies, we have formulated an updated definition of corporate sustainability for mining companies. Corporate sustainability for a mining company is its ability to identify ESG risks and other types of risks, manage them in the short term, and create conditions for corporate sustainable development (CSD) by maintaining and building up the resource potential, using natural assets, and implementing circular and CSR strategies that reflect the interests of the company's stakeholders and are adapted to the environment in which the company operates. CS and CSD ensure the creation of the company's value, which is understood as market capitalization factoring in ESG limitations in the short and long terms. The company's involvement in the key SD issues at the meso and macro levels ensures that the balanced interests and needs of the key stakeholder groups are met through ESG strategies, CSR implementation, and the choice of circular business models that ensure CS and CSD.

Our annual report analysis of whether it is possible to manage CS with a view to creating competitive advantages in coal companies showed the following. In its 2020 annual report, SUEK, the largest Russian company, reveals the need to revise risk management approaches at all management levels based on ISO 31000 and COSO ERM standards. This is in line with today's ESG requirements [110] and ensures the coal company's CS.

The most important factor in improving the CS of mining companies may be the integration into corporate strategies of circular business models related to the reduction of mining waste and making the supply chains of secondary raw materials circular, as well as the decarbonization of energy supply for the main and auxiliary production processes.

Thus, within the framework of SUEK's environmental strategy for 2018–2020, there was an increase in the share of used and recycled waste by 20%, a decrease in water consumption per unit of electricity produced by 14.5%, and an increase in water reuse by 10%.

In 2020, the company developed and implemented an innovative technology for the processing of sludge coming from the operating processing plant and from old sludge sedimentation tanks. The additional output of marketable products (150,000 tons per year) created economic effects, and there were decreases in the amounts of processing waste, emissions, dust, and noise caused by transporting waste by road.

An example of SUEK's work in the area of decarbonization is the project to transfer the heat load from stand-alone boiler houses to thermal power plants and reduce the carbon footprint due to heat and electricity cogeneration, which reduces $CO_2$ emissions per unit of energy produced by 32% due to the improved efficiency [150]. As a result of the measures taken to increase resource conservation in 2020, 4.8 million m3 of methane was utilized, and emissions decreased by 0.115 million tons of $CO_2$ equivalent due to the replacement of old boiler houses. This indicates that the company is aligning its activities with the ESG requirements for the SDGs, ensuring CS and creating the foundation for CSD when implementing circular business models.

## 5. Conclusions

So far, academic studies have not developed a universal theory of corporate sustainability. There are several different concepts:

- Those focused on SD and projected at the micro level;
- Those focused on the stakeholder concept in a broad sense;
- Those reflecting the relationship between CS and CSR;
- Those aimed at the application of strategic management for the purposes of CS.

Corporate sustainability of a mining company is its ability to identify ESG risks and other types of risks, manage them in the short term, and create conditions for corporate sustainable development (CSD) by maintaining and building up the resource potential, using natural assets, and implementing circular and CSR strategies that reflect the interests of the company's stakeholders and are adapted to the environment in which the company operates.

CS studies are not focused enough on analyzing the distinguishing features of individual industries, which are mainly taken into account when assessing CS, for example, in SASB standards.

There is an ongoing discussion in academic literature on whether the features of the mining industry are important in assessing CS. There are two opposite points of view: some researchers claim that the mining industry has its distinguishing features, while others argue that it does not.

The need to improve the CS of mining companies is due to the growing environmental risks, their significant socioeconomic impact on the mining regions, and the presence of additional industry-related regulations at the national and international levels.

Natural capital determines the key specific features of the mining industry. The management of natural capital is indirectly included in the company's management process, requiring special tools to assess and improve CS.

The relationship between sustainability and CE is not always clear due to differences in the conceptualization of CE, which creates obstacles to circular ideas and models in corporate governance in those sectors which regulators do not yet prioritize in the transition to CE, including the mining sector.

The concepts of CE, sustainability, and SD at the micro level are consistent in their basic interpretations. The inclusion of circularity indicators in sustainability reporting in the ESG context is likely to act as a catalyst of the adoption of CE principles and the development of circular business models.

A factor in improving the CS of mining companies may and should be the integration into corporate strategies of circular business models related to the reduction of mining waste and making the supply chains of secondary raw materials circular, as well as the decarbonization of energy supply for the main and auxiliary production processes.

## 6. Future Research and Limitations of the Study

This study contributes to creating a broader picture of corporate sustainability that includes mining companies focused on adding CE elements to their practices in order to gain long-term competitive advantages in the context of the green transformation.

We propose to include the circularity indicator in the corporate ESG reporting of mining companies, which will become an additional incentive for creating partner ecosystems based on the CE elements that support sustainable development. This requires further research to develop appropriate indicators and evaluation models. One of the most difficult aspects is assessing the risks of transitioning to circular mining business models in the face of uncertainty and assessing the chances of a successful transition.

Regarding the limitations of the study, we studied how using CE practices in the core activity of a mining company contributes to achieving SDG 12, but we did not consider SDG 7, SDG 9, SDG 6, SDG 8, and SDG 11, whose achievement can be greatly promoted by following CE principles [151].

In exploring the relationship between corporate sustainability and circularity, which largely depends on the institutional environment and CE policies, we touched upon the experience of China but did not discuss that of the EU, a leader in CE trends. This is explained by the fact that, unlike the EU, China covers the mining industry as one of the national priorities in its CE regulation. While calling for the promotion of best practices in mine waste management, neither the EU Action Plan for Circular Economy (COM(2015) 614 final) nor the new Circular Economy Action Plan (2020) [152] considers mining a priority sector for a circular economy.

The analysis carried out by the authors identified the following areas for future research:

- The identification of the opportunities and risks of using CE elements at the micro level in order to increase the CS and positively influence the CS of mining companies. As the main elements, circular business models, circular R strategies, and the possibility of influence through the circularity indicators included in the ESG reporting of companies can be considered.
- The determination of the possibility of using ESG metrics for CS assessment and determination of CG management tools in mining companies.
- The determination of the impact of competitive advantage on corporate sustainability when substantiating CSD strategies in mining companies.

**Author Contributions:** Conceptualization, T.P. and E.B.; methodology, V.K., T.P. and E.B.; validation, T.P. and V.K.; formal analysis, V.K.; data curation, E.B.; writing—original draft preparation, T.P., V.K., E.B.; writing—review and editing, T.P., V.K., E.B.; visualization, E.B. All authors have read and agreed to the published version of the manuscript.

**Funding:** This research received no external funding.

**Institutional Review Board Statement:** Not applicable.

**Informed Consent Statement:** Not applicable.

**Data Availability Statement:** Not applicable.

**Conflicts of Interest:** The authors declare no conflict of interest.

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
