# Peer review of "Analyzing the Concept of Corporate Sustainability in the Context of Sustainable Business Development in the Mining Sector with Elements of Circular Economy"

_sustainability, doi:10.3390/su14138163_

Round 1

Reviewer 1 Report

The article "Analyzing the concept of corporate sustainability in the context of sustainable business development in the mining sector with elements of circular economy" by Ekaterina Blinova, Tatyana Ponomarenko, Valentin Knysh is submitted for review.

This article reveals a rather relevant topic: the sustainable development of countries or regions with a raw-material orientation of the economy. The relevance of the chosen topic is well and voluminously substantiated. A good review of previous studies has been made. However, from my point of view, there are a number of minor comments, the correction of which will improve the quality of the manuscript and improve the perception of these studies by the reader.

1) Narrow geography of cited studies. In the conduct, the authors refer mainly to the research of Russian scientists.

2) Despite the fairly well-described topicality (introduction), a broad review of the literature (Chapter 3), the authors touch upon a rather narrow sector of the resource-based economy - coal. My opinion is based on Fig. 1. It indicates the indicators of the coal industry only. From my point of view, it is necessary to present indicators for other sectors of the mining and oil industry.

3) Also, the authors do not provide environmental indicators for various countries, which also reduces the value of this manuscript. In this situation, I would like to recommend adding information on other industries or slightly changing the name, that is, indicating in it the resource-based economy under study, adding “on the example of the Russian coal industry” to the name. This statement also follows from remark 1.

Remarks 2 and 3 are advisory in nature. If the authors consider that making such changes is inappropriate and this may overload the article. I have no objection to the publication of the manuscript without these comments.

4) With a fairly large amount of citation, I am surprised by the following fact: in the paragraph (lines 134 -137) they talk about quite a serious impact of the mining industry on the environment. However, only the work that discloses "sustainable management of transboundary groundwater resources" is cited. It seems to me that this link is forced and cannot reveal the full seriousness of environmental risks.

5) The same applies to the wording made by the authors in line 137-140. They argue about accelerating the greenhouse effect or reducing oxygen production due to deforestation and the inability to use drinking water. When they refer to the work that explores the "influence of coal companies on the socio-economic development of coal-mining regions"

In the case of remarks 4 and 5, I see a clear need to refer to some studies of scientists that concretize the concept: "heavy impact on the natural environment". For example, in the works of the Russian scientist V.I. Golik conducted many studies on the impact of the mining industry on the ecology of the region (for example: "Protection of natural geological environment by utilizing ore tailings" and others). Many works are devoted to environmental risks in the regions of the mining industry. In the work "Utilization of Mineral Waste: A Method for Expanding the Mineral Resource Base of a Mining and Smelting Company" all forms of the impact of the mining industry on the environment are widely covered, a classification feature is defined, a classification is made and highlighted: groups of violations; types of violations and characteristics of violations. In the work "Creation of backfill materials based on industrial waste", using a mathematical model that takes into account the degradation of the ecosystem, a reserve of technogenic damage is determined. I only pay attention to one paragraph, but the list goes on. It seems to me that with a sufficiently large amount of references, they do not fully reflect and do not prove the statements of the authors.

From my point of view, the References should be cautiously revised to eliminate “forced” references and add sources that confirm the claims of the authors.

6) According to the authors, with which I fully agree, sustainable development issues have been widely discussed recently. However, the authors did not cite previous studies to determine the likelihood of man-made disasters, the cumulative multiplier effect, taking into account environmental degradation and a number of others, which will allow more accurately and clearly solve the tasks and achieve the goal of the study.

Sincerely

Author Response

We are very grateful for receiving the article review. We have received many helpful suggestions to improve our article and strengthen the material in some areas. Comments have been taken into account.  Unfortunately, none of the authors can see the line numbers in the document, so we wrote the sections with changes.We have attached the full pesponses.

Reviewer 2 Report

Corporate sustainability in mining refers to corporations delivering minerals in a sustainable manner. This research takes Russia as an example to analyze the influence of the mining industry on the environmental, economic, and social development of both a country with a resource-based economy and individual mining regions. Overall, the manuscript is well-written. However, it can also be further improved based on the following suggestions.

 1)      Line 38: Define SD on first use. The abstract should stand alone. Therefore, you should follow the trend of defining abbreviations in the rest of the manuscript too.

 2)      Line 92-95: After this statement, the authors should give a review of the literature on sustainable mining/green and climate-smart mining, which address circular economy approaches in the mining sector. For instance, you may review these articles: https://doi.org/10.1016/j.jclepro.2022.132055 ; https://doi.org/10.1016/j.eswa.2021.116062; https://doi.org/10.1016/j.resourpol.2021.102007.

 3)      The goal and objectives of the study can be better written as a paragraph.

 4)      The section “Materials and Methods” is not up to a satisfactory level since neither the materials nor the methods are appropriately described within the manuscript. In particular, the methodology is poorly articulated. There is nothing to be considered a significant novelty in the method, at least not in the way it is given in the manuscript.

 5)      The Results and Discussion section is not coherent and comprehendible since it appears hastily written. The paragraphs are too long and there is lack of figures/tables that constitute work outputs.  

Author Response

We are very grateful for receiving the article review. We have received many helpful suggestions to improve our article and strengthen the material in some areas. Comments have been taken into account.  Unfortunately, none of the authors can see the line numbers in the document, so we wrote the sections with changes. We have attached the full pesponses.

Author Response

(The authors gave the same response as above.)

Round 2

Reviewer 1 Report

Dear Authors 

You responded to most of my comments and I respect your answers even if I'd still support my position. Diversity of view-points is a value in science.

The work was significantly improved following all Reviewers' comments. I opt for acceptance in present form. Just one editorial issue that should still be considered. I'd recommended to mark clearly sources that available in Russian only. It is not mandatory but might be helpful for the Readers.

Sincerely

Author Response

Thanks for the recommendation! We added "In Russ." to each reference in Russian.

Reviewer 2 Report

The revision is satisfactory.

Author Response

Thanks for the recommendations! We have analyzed the recommended topics and articles and have added information (lines 1036-1059).

This manuscript is a resubmission of an earlier submission. The following is a list of the peer review reports and author responses from that submission.

Round 1

Reviewer 1 Report

This is an interesting study - below a few points to further improve this work:
- the abstract is usually considered as a stand-alone part of the paper - meaning that you need to define the abbreviations again when they appear in the text, see SD, etc.
- I really like how you explain the objectives of this study in line 191-198 - after that there is big text block from line 217-479 - that is very comprehensive - maybe try to chop this into smaller parts that are directly linked to the objectives of this work
- line 983 - "companies"
I feel this work fits well into the scope of Sustainability - maybe - if possible try to provide more quantitative analysis in the abstract and conclusions - given the topic of this work I understand that this is hard - but maybe you can link it better to the number of sources reviewed - just a suggestion